# Comparative Analysis of the Implementation of Support Vector Machines and Long Short-Term Memory Artificial Neural Networks in Municipal Solid Waste Management Models in Megacities

**DOI:** 10.3390/ijerph20054256

**Published:** 2023-02-27

**Authors:** Johanna Karina Solano Meza, David Orjuela Yepes, Javier Rodrigo-Ilarri, María-Elena Rodrigo-Clavero

**Affiliations:** 1Department of Environmental Engineering, Santo Tomás University, Road 9 Street 51-11, Bogotá 110231, Colombia; 2Instituto de Ingeniería del Agua y Medio Ambiente (IIAMA), Universitat Politècnica de València, 46022 Valencia, Spain

**Keywords:** artificial neural networks, municipal solid waste, support vector machines, solid waste management, waste disposal

## Abstract

The development of methodologies to support decision-making in municipal solid waste (MSW) management processes is of great interest for municipal administrations. Artificial intelligence (AI) techniques provide multiple tools for designing algorithms to objectively analyze data while creating highly precise models. Support vector machines and neuronal networks are formed by AI applications offering optimization solutions at different managing stages. In this paper, an implementation and comparison of the results obtained by two AI methods on a solid waste management problem is shown. Support vector machine (SVM) and long short-term memory (LSTM) network techniques have been used. The implementation of LSTM took into account different configurations, temporal filtering and annual calculations of solid waste collection periods. Results show that the SVM method properly fits selected data and yields consistent regression curves, even with very limited training data, leading to more accurate results than those obtained by the LSTM method.

## 1. Introduction

The increasing generation of municipal solid waste (MSW) and the need for its proper management is one of the main environmental problems that must be addressed in large urban areas [1]. Municipal solid waste management (MSWM) requires alternatives that optimize every stage within the process. A critical concern is the lack of control and poor management of waste generated in urban centers, which hinders its sustainable management, treatment, collection and final disposal [2]. Adequate and sustainable waste management infrastructure planning depends on the ability to reliably estimate future MSW generation. This task requires the consideration of expected demographic, social and economic factor changes. Therefore, performing accurate MSW forecasting is complex and challenging [3,4].

Currently, efforts have been made to transform the waste management industry toward sustainability and profitability through the implementation of advanced technologies and smart systems [5]. Recent developments in new software and internet technologies, together with a gradual introduction of more compact and reliable hardware products, have demonstrated the ability to precisely manage MSW procedures with greater ease than costly and tedious field experiments [6]. Artificial intelligence (AI) effectively provides these systems with alternatives to minimize problems associated with decision making and data management for such specific activities and processes.

AI has been widely implemented in the environmental engineering field to solve problems related to air pollution, water and waste-water treatments, soil remediation, groundwater contamination and planning MSW strategies [7], among other activities. These tools are necessary in order to optimize MSW activities, analyzing all the available data and leading to environmental, social and economic benefits. Modeling and simulation tools must be implemented to establish relationships between the different variables involved.

AI is widely used to forecast MSW generation, classification, properties and collection patterns, situate facilities and simulate waste conversion processes, among other activities [8]. MSW generation analysis is a crucial stage when designing an efficient MSW management system. Accurately forecasting MSW generation depends on the existence and availability of accurate data [9]. When these data are not available, modeling methods are needed to predict MSW generation. The success of these modeling techniques largely depends on the selection of waste flows [3]. Traditional prediction of MSW generation has been supported with the application of different forecasting tools such as descriptive statistical analysis, regression analysis, principal component analysis, time series analysis and material flow analysis [8,10,11].

The AI systems most frequently used for modeling and optimizing MSW processes include artificial neural networks (ANNs), support vector machines (SVMs), linear regression (LR), decision trees (DTs) and genetic algorithms (GAs) [5,12,13]. Neural network tools are the most widely used methods in this field, including radial basis function (RBF), multilayer perceptron (MLP), back propagation (BP) and feedforward, autoregressive and recurrent ANNs [5,12,13]. Other less frequently used models are adaptive neuro-fuzzy inference systems (ANFISs), random forests (RFs), wavelet transform (WT), K-means, data mining, Naïve Bayes, rough sets, logistic model trees, Q-type clustering, ant colony optimization, non-inferior set estimation (NISE), goal programming and artificial immune systems (AISs) [5].

ANNs are effective in modeling processes with sets of incomplete or uncertain data, as well as in addressing complex or imprecise problems that require human intuition [14]. They have been successfully applied in waste prediction and generation problems, waste classification, biogas generation, leachate formation, energy recovery, waste heating value, determination of the melting temperature of waste and the design of optimal waste collection routes [3,5]. ANNs are one of the most popular non-linear models and have been successfully applied in predicting the production of municipal solid waste [15,16].

ANNs have been widely used to model various MSW processes due to their robustness, fault tolerance and suitability for representing complex relationships among variables in multivariate systems. Furthermore, the calibration process of ANN systems usually requires fewer parameters than those required by deterministic models [17,18]. On the other hand, ANNs are ill-suited for handling logical and arithmetic problems that need a high degree of precision, as they are prone to overfitting [5]. Moreover, they are incapable of determining the relative importance of several factors involved in an analysis [5].

There are a wide range of ANNs with different purposes and diverse architectures based on their established prediction tasks. Regarding time series, LSTM recurrent neural networks have led to favorable results and the design of their architecture is useful in exploring temporal correlations among data. In general, algorithms based on neural networks learn non-linear and non-parametric functions based on a set of training data.

Support vector machines (SVMs) are supervised machine learning algorithms that are useful in data analysis [19,20]. These tools are binary linear classification techniques, which separate classes with the largest gaps between border line instances. For non-linearly separable data problems, SVMs have been extended using kernels, mathematical functions that transform the data from a given space to a new high-dimensional space where data can be separated with a linear surface [21].

Despite being initially applied to address classification problems, SVMs have been used to solve regression problems as well, outperforming several classical regression techniques [5]. These algorithms are less susceptible to overfitting and perform at an expert level in reducing error estimates and model dimensions, unlike statistical procedures such as principal component analysis (PCA), which only address the dimensionality of the model [22]. This tool has been used in several areas of environmental engineering, such as air pollution modelling [23], the design of systems used to identify plastic solid waste (PSW) based on near-infrared reflectance (NIR) spectroscopy in combination with SVMs [24], forecasting the generation of photovoltaic energy [25], analyzing daily global solar radiation models and their respective comparison [26], and predicting the biodegradability of organic chemicals [27].

Some applications of SVM tools to MSW management problems show that they have been widely used to forecast container filling levels, waste generation, waste classification, energy recovery, and waste heating value [5,8,13]. SVMs are often improved to develop models in combination with a principal component analysis (PCA) technique to forecast waste production [28].

Therefore, the application of AI techniques in the design of MSWM systems, especially in the evaluation of waste production, is a topic of special interest. In this work, the results obtained when using SVM and LSTM techniques are compared. This paper is organized in six sections. Section 1 includes relevant references about the use of AI techniques in MSWM systems. Section 2 summarizes the basic concepts of the SVM and LSTM algorithms, the description of the study area and the characteristics of the methodological modeling process. Section 3 shows the results obtained when applying SVM and LSTM techniques to the available data. Section 4 presents a discussion of the results and chapter 5 includes the main conclusions.

## 2. Materials and Methods

ANNs and SVMs are appropriate AI tools to model processes related to solid waste management. This work shows a comparative analysis of the performance of these tools on an MSW management model including socio-economic variables. The case study is of the megacity of Bogotá (Colombia). MSW management in Bogotá is a complex problem due to the large amount of waste generated by the megacity’s population, which is finally disposed of at the Doña Juana landfill site.

Several AI software platforms and libraries have been used to solve MSW management problems. MATLAB was used in the training and testing of neural networks and MLP algorithms [29,30,31,32,33,34,35]. SPSS was used to correlate model attributes such as population density and waste generation [17,36,37,38]. R software was used to remove outliers from data sets, particularly in waste generation simulations [39,40]. C++ and Python languages were used to simulate the AI models in the same way [41,42,43]. C++ provides different AI libraries that include OpenNN, OpenCV, BOOST, gflags, glog and Tensor Flow. Similarly, Python has Tensor Flow along with other libraries such as Spyder, Matplotlib and Utils [5].

To perform a comparative analysis, the same variables in the two models were used: population, solid waste production, socio-economic stratification, transportation service expenses and MSW disposal costs using different technologies. Nine analysis scenarios were integrated into each model, simulating the behavior of each scenario in a given year with both LSTM and SVMs. The simulation process was designed to study the behavior of highly complex cities such as the megacity of Bogotá. Smaller areas can be considered in order to carry out specific analysis according to the particularity of each zone. The results explained below provide an analysis of alternatives to support objective decision making in MSW management, showing the outcomes of implementing AI tools based on ANN and SVM techniques.

### 2.1. Basic Concepts of the Algorithms Used in this Study

#### 2.1.1. Artificial Neural Network (ANN) and Long Short-Term Memory Network (LSTM)

ANN theory was first proposed by McCulloch and Pitts in 1943 [44]. ANN uses mathematical modelling to simulate some structures and functions found in the neuronal system of the human brain [45]. The principle of ANNs provides them with a good nonlinear mapping capability and makes them suitable for solving a mapping problem from one data set to another [12]. In accordance with signal transmission modes, ANNs can be classified into feedforward and feedback neural networks. Feedforward neural networks are relatively simpler and widely used in MSW management studies [7].

In a multilayer feedforward neural network, neurons in each layer connect to neurons in the next layer through links of different weights (Figure 1). Neuron layers can be classified into three types: input layer, hidden layers and output layer. The neurons in the input layer form a system that receives external information, such as sensory receptors; the hidden layer neurons simulate a biological neural network to transmit information; and neurons in the output layer present decision output [12].

The mathematical description of an ANN can be understood by Equation (1) [46].
Y(t) = F (∑_i=1_
^n^(X_i_(t) W_ij_(t) + b_ij_))(1)
where:X_i_(t) is the input value at time tW_ij_(t) is the weight of neural input at time tb_ij_ is the biasF is a transfer functiony(t) is the output value at time t

Recurrent neural networks (RNNs) are variants of neural networks that are good at dealing with sequential data processing [47]. The structure of the ANN is organized iteratively, such that output data are converted to input data taking into account the stored output of the previous time step t − 1, which is added to the inputs of the current time step t. This configuration means that a change in the state of an individual neuron can be transferred via feedback to the other neurons, invoking transient states and generally leading to another state of the network [48].

LSTM is meant to solve the challenges faced by RNNs with the help of gates that manage the flow of sequences at the current state and output of the current sequence [49]. The idea of LSTM is that it maintains the state of the memory for a long time due to the presence of the memory cell. The memory state consists of gates that regulate data flow in the memory. The memory state is present in all LSTM cells to modify the information values of the previous states based on the importance of the gate units [50].

LSTM networks are composed of a sigmoid neural network layer and a point multiplication operation. To avoid having to concatenate vectors at the output of a step only to later separate them at the beginning of the next step, LSTM cells are usually represented with three inputs [51]. Figure 2 shows a representation of the LSTM model and its mathematical formulation can be found in [52].

LSTM has advantages over other deep learning algorithms: it learns data behavior better than RNNs from any predetermined value [49] and generally requires a minimum number of hyperparameters for fine tuning [53].

#### 2.1.2. Support Vector Machines (SVMs)

In the 1990s, Vanik systematically introduced statistical learning theory and proposed the SVM algorithm [54]. Due to its excellent performance in the field of text mining and error diagnosis, SVM gradually became the mainstream technology of machine learning methods [55]. It is used in solving classification and regression problems, being a linear model that provides solutions for both linear and non-linear problems [56].

SVM is a probabilistic-based technique that performs binary classification and aims to find the dividing hyperplane that separates both classes of the training set with the maximum margin [57]. Any training samples that fall on hyperplanes H_1_ or H_2_, the sides defining the margin, are support vectors, as shown in Figure 3. The equations that solve the LSTM model can be found in [58].

The SVM algorithm has many advantages: it works skillfully even with semi-structured and unstructured data, it can handle any complex problem with the proper function, it can work well with high-dimensional data and due to generalization, it has less risk of overfitting. The main disadvantage is that it requires more time to train the model for a large data set and that it does not work well with noisy data [56].

### 2.2. Description of the Study Area

According to 2020 projections, Bogotá’s population is 8,380,801 inhabitants [59], and 196,138 tons of solid waste were collected in March 2020 [60], which were disposed at the Doña Juana MSW landfill. This huge MSW production underscores the capacity required by the waste treatment systems of the city. Bogotá is administratively subdivided in the following 20 localities: Usaquén, Chapinero, Santa Fe, San Cristóbal, Usme, Tunjuelito, Bosa, Kennedy, Fontibón, Engativá, Suba, Barrios Unidos, Teusaquillo, Los Mártires, Antonio Nariño, Puente Aranda, La Candelaria, Rafael Uribe, Ciudad Bolívar and Sumapaz [61]. The megacity has a waste collection system divided into five ESAs (exclusive service areas). Currently, each ESA is operated by a specific company that provides waste collection services [62] (Figure 4, Table 1).

Doña Juana landfill is Bogotá’s primary site for the final disposal of waste. Its presence is vital for the city’s development and it is located in Ciudad Bolívar [64]. Given the quantity of waste collected in the city and finally disposed of at this landfill site, health emergencies are common there, demonstrating the need to propose alternatives that minimize the amount of waste stored at the landfill.

### 2.3. Methodological Model

As stated above, the main objective of this work was to carry out a comparative analysis of the performance of the LSTM and SVM models to examine the precision of predictive learning models and estimate the related costs of each. The development stages of this research were as follows:Data review and standardization;Forecast of urban solid waste generation performed by LSTM and SVM;Proposal of scenarios for methodological development: E1–E9;Inclusion of transportation and treatment costs of solid waste for each scenario;Modeling the developed methodology integrated into LSTM and SVM;Results analysis.

As a first stage of this study the values of the following model parameters were collected to estimate the waste production:the city’s population;solid waste generation by collection area;the city’s socio-economic stratification;transportation expenses;possible waste treatment costs.

Population data were obtained from information provided by the National Administrative Department of Statistics (DANE) and the District Planning Secretariat [65]. The values of the MSW per capita production in 2016 were based on data provided by the Special Administrative Unit for Public Services [66]. In this work, socio-economic stratification is understood as “the classification of residential buildings in a municipality, which is performed based on the Regime of Public Home Services in Colombia” (Law 142 of 1994). Strata 1, 2 and 3 correspond to the lowest strata areas (fewer economic resources), while strata 5 and 6 correspond to highest strata areas (more financial resources). The location of these areas and the strata distribution data were provided by DANE [67]. Once these data were collected, projections in time were carried out by implementing LSTM for each collection area while simultaneously carrying out the same process for the SVM models.

A set of scenarios to be evaluated were then determined (Table 2). To this end, the same scenarios proposed in the “Study of Alternative Techniques for the Treatment, Final Disposal and/or Use of Solid Wastes—Proposed Adjustment to Decree 838 of 2005 (Compiled in Decree 1077 of 2015)” by the Inter-American Development Bank were used [68]. This study analyzed the treatment and disposal costs of a city’s solid waste under different alternatives. The possible revenue obtained from byproducts in each scenario was also calculated, considering the existence of three differently sized waste treatment facilities in each scenario.

Waste transportation costs were included within the model, following the requirements established by Resolution CRA 853 of 2018 of the Commission on the Regulation of Drinking Water and Basic Sanitation [69]. This resolution determines the maximum costs established for 2018, which depend on:-the monthly average of tons collected and transported in the immediately preceding year (tons/month), as well as-the distance to the final disposal site, transfer station or treatment plant (km).

To update costs, [69] also establishes an increase factor from the Consumer Price Index (CPI) in the month selected by the provider to be used as the basis for the update in addition to the corresponding factor for the last month in which the CPI was updated [69]. 

Given the relationship between distance and transportation costs, the distances from the ESAs’ geographic mass centers to the most likely treatment sites were used as the basis for the corresponding calculations, taking into account areas in the city that have the possibility of obtaining environmental licensing for this type of process. These geographic mass centers are included in Decree 652 of 2018 of the Office of the Mayor of Bogotá [70]. Adjustments were made for the final calculation of all the costs in the same manner as those associated with waste treatment, using the updating factor from 2015 to 2020 and applying the present value concept to 2020 in the predictive model. 

Therefore, both LSTM and SVM models were used to simulate the same set of data and to obtain results for the same set of scenarios. This research allows us to identify which is the best scenario for the megacity of Bogotá while evaluating the performance of the two AI tools used. 

In both models, Python was used as the programming language via Jupyter Notebook in the Google Colab environment.

## 3. Results

Results of the analyzed models for each of the city’s solid waste collection areas (ESAs) are presented below. SVM results were obtained using radial basis kernel functions. The SVM model provides good adjustment to the training data and replication of their variability pattern with time while LSTM learns a more general pattern and results in noticeable periodic predictions. Figure 5 shows results obtained for ESA 1 for both methods. 

The network architecture was structured with one input neuron, ten hidden neurons and one output neuron. A hyperbolic tangent function was used as the activation function and a linear function was used for the dense layer.

It is important to note that SVMs capture the growth pattern in the training data as the predominant behavior in the data for its subsequent replication in the forecasts. These effects were clearly seen in ESA 2 (Figure 6).

A similar behavior to that observed for ESA 1 was seen for ESA 3. In this zone SVMs adjust more accurately to the training data, generating more variable predictions than those simulated by LSTM (Figure 7).

Figure 8 and Figure 9 show the predictive behavior of ESAs 4 and 5, respectively. For these ESAs, the best predictions are obtained through SVM implementation, maintaining the trend of the results obtained for the prior ESAs.

To quantify the error associated with each prediction made by these methods, the mean squared error (MSE) and the mean absolute error (MAE) were used. Table 3 shows the results obtained for each ESA by both modeling techniques. 

The results shown in Table 3 highlight that the SVM method provides simulations with fewer training errors. Moreover, ESAs 2, 4 and 5 show the highest absolute errors in the training using the SVM method. For LSTM, the highest absolute error values were obtained in ESAs 3 and 4. The lowest value for the MAE is for ESA 1, both for SVM and LSTM simulation methods.

From an economic perspective, treatment and transportation costs for 2030 within the models were given in present 2020 values. As mentioned above, for cost and revenue projections, this study used scenarios proposed in the “Study of Alternative Techniques for the Treatment, Final Disposal and/or Use of Solid Wastes—Proposed Adjustment to Decree 838 of 2005 (Compiled in Decree 1077 of 2015)” by the Inter-American Development Bank [67].

Two different scenarios were considered in the modeling process.
Scenario 1—The waste is disposed of at the Doña Juana landfill as is currently carried out (Distance 1). This scenario considers the potential revenue obtained from the sale of byproducts from the primary treatment process;Scenario 2—The waste is disposed of at a second landfill (Nuevo Mondoñedo), located at Distance 2. This landfill site is still not licensed and depends on the results of feasibility and environmental studies (Figure 10).

Scenario 2 was used to carry out a comparative cost analysis between ESAs and the models, based on the change in distances (Table 4). In addition, the comparison of results between both scenarios also demonstrates the model’s flexibility, which allows for future adjustments as more data are obtained and different MSW treatment techniques are considered in each ESA.

Figure 11 shows the comparative results in terms of ESA-related costs, models and distances.

ESA 2 shows the highest values in the cost analysis as it is the location that generates the highest production of MSW in the city. This production is proportional to the population density in that area. The lowest values were found in ESAs 4 and 5, as they correspond to the zones with the lowest population densities in the city. 

Scenario E6 shows the lowest costs for every area, using either SVMs or LSTM. This scenario considers treating 60% of the waste for composting, while 40% is disposed of at the landfill. 

ESA 4 shows the lowest costs. The monthly average cost was lower for scenario E7 (USD 2,326,874). However, if the possible revenue generated in the process is considered, the best scenario would be E6 (source classification + composting + landfilling), with a monthly average value of USD 16,435,044. These results were obtained by modeling with SVMs and considering the Doña Juana landfill as a final disposal site. 

As ESA 2 shows the highest costs, its most favorable scenario is E7, with costs equal to only USD 6,446,908 as a reference. If the projected revenues for the scenarios are included in the calculations, scenario E6 is the best, with a monthly average value of USD 4,628,449. This pattern is consistent for all the ESAs.

## 4. Discussion

Results obtained in this study demonstrate the applicability of implementing models that use LSTM and SVMs to forecast the generation of urban solid waste, as well as to calculate potential long-term costs. Taking into account the characteristics of the processes associated with MSW management, the chances of managing large volumes of data are scarce. This limitation constrains the possibilities of modeling with these types of tools over the medium and long term. Insufficient data is a major obstacle that affects the implementation of AI systems. 

AI models are primarily based on large data sets for training and calibration purposes. Lack of data often occurs with MSW due to missing or incomplete residual data; this is partly the result of the majority of MSW industries being outdated, with limited reliable records and scarce sensory data, especially in developing countries [5]. However, the models presented in this work are flexible and allow for data to be continuously updated, which makes it possible to minimize errors and obtain better results for a given period of time.

The numerous AI models and their rapid evolution distract from efforts to incorporate AI in MSW. There are a great deal of models, each one reporting successful results when compared to conventional methods. However, the overall progress does not seem to be as significant as expected, given the number of studies [5,8,12]. That said, studies often set out to perform comparative analyses between tools such as behavior analyses between ANNs and decision trees, which include socio-economic variables, with the ANNs obtaining better results [72]. Similar results were obtained for MSW predictive analyses by comparing the two tools and including support vector machines in the analysis. SVMs showed better results, followed by the ANNs and, lastly, decision tree algorithms [73]. In-depth studies have compared a greater number of algorithms, such as smart systems that include SVMs, adaptive neuro-fuzzy inference systems (ANFISs), ANNs and k-nearest neighbors (kNNs), to determine their ability in forecasting monthly waste generation. Results show that AI models have sound predictive performance and could be successfully applied to establish municipal solid waste projections. In this case, the ANFIS and kNN models showed the best performance [74].

Following the trends in this line of research, a comparative analysis of two models to carry out predictive analyses and projections on treatment cost calculations has been shown above. The first model used an ANN, while the second used SVMs. The decision to use LSTM in this work was based on its direct application to time series prediction. One of the main advantages of these networks is their ability to adjust non-linear data behavior and maintain memory and forget states which take into account past time information. Moreover, the primary advantage of SVMs is how they properly adjust to the data despite its variable nature, or when faced with problems with a small amount of training data. Moreover, using different kernels is a possibility, as they are better-suited for interpreting training data for improved forecasting. 

The analysis carried out in this study included the estimation of the economic costs for every scenario and every year over the first 3 years of forecasts for ESA 2, ESA 4 and ESA 5, in which the costs vary according to the patterns learned in prior years. However, when trying to consider predictions for the distant future (more than 5 years), costs tend to average out historically. This observation may be explained by the fact that weak patterns learned in these areas are likely to disappear due to errors propagated in distant future forecasts.

## 5. Conclusions

Results obtained in this study show that SVMs provided the best adjustment to observation data. SVM simulations properly adjust to the data and achieve coherent regression curves, despite having very limited training data. However, their behavior was not suitable in every zone. In ESA 1 and ESA 3, SVMs apparently showed prediction patterns based on the series seen in the training, but in ESA 2, ESA 4 and ESA 5, results were not close to the actual observation data. 

LSTM’s best predictions were obtained for ESA 1 and ESA 3. However, LSTM’s performance was generally overshadowed by results obtained using SVMs. SVM simulations show the lowest training error values. ESA 2, ESA 4 and ESA 5 had the highest absolute error values in training, which may suggest that the data for these series do not follow patterns that are easy to model, and require more sophisticated pre-processing strategies, in addition to more comprehensive model parameter adjustments.

Scenario E7 (landfill + biogas capture and burning) showed the lowest costs for MSW treatment. However, if the possible revenue from the sale of byproducts is considered, the most convenient scenario is E6 (source classification + composting + landfilling), using either SVMs or LSTM. Scenario E6 allocates 60% of the waste for composting, while leaving 40% to be disposed of at the landfill. 

Results show the relevance of implementing selective source classification systems at the source by having specific containers for each byproduct that is part of the mixture of municipal waste.

## Figures and Tables

**Figure 1 ijerph-20-04256-f001:**
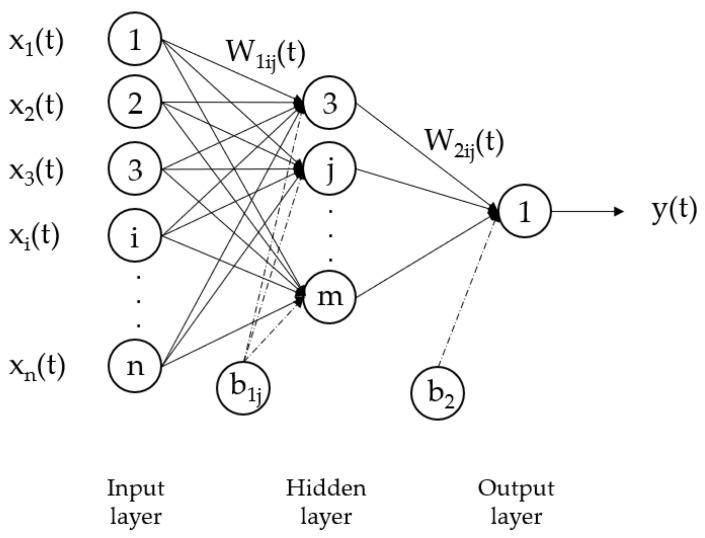
Structure of a multilayer feedforward ANN.

**Figure 2 ijerph-20-04256-f002:**
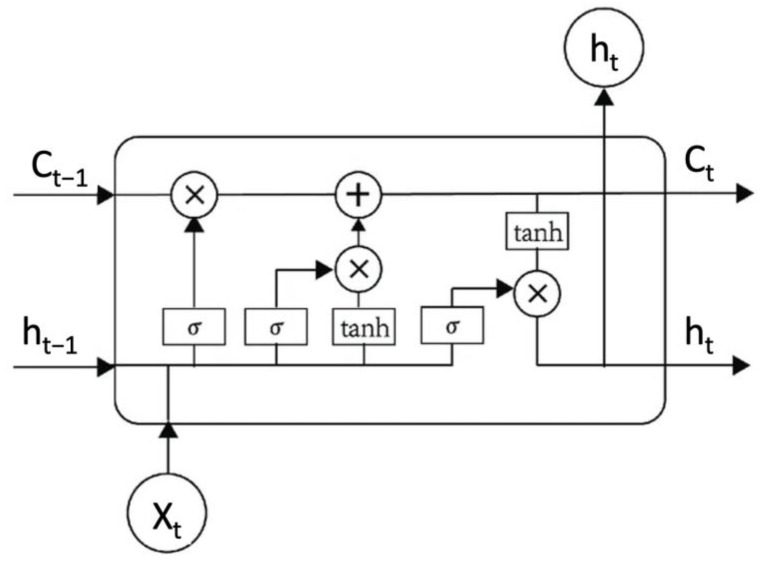
LSTM model representation.

**Figure 3 ijerph-20-04256-f003:**
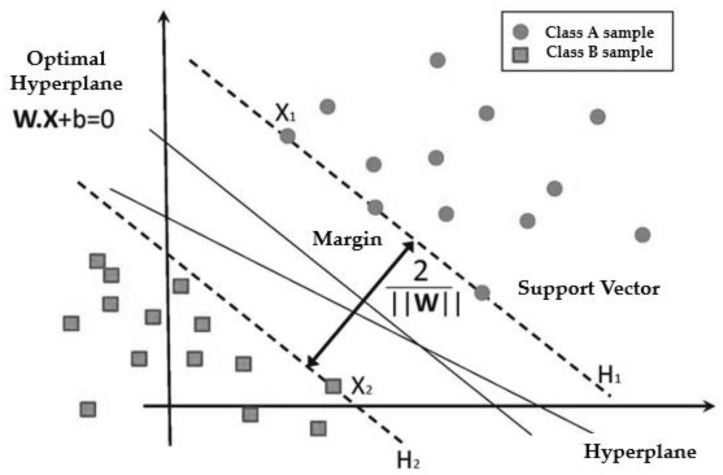
Classification of data by support vector machine (SVM).

**Figure 4 ijerph-20-04256-f004:**
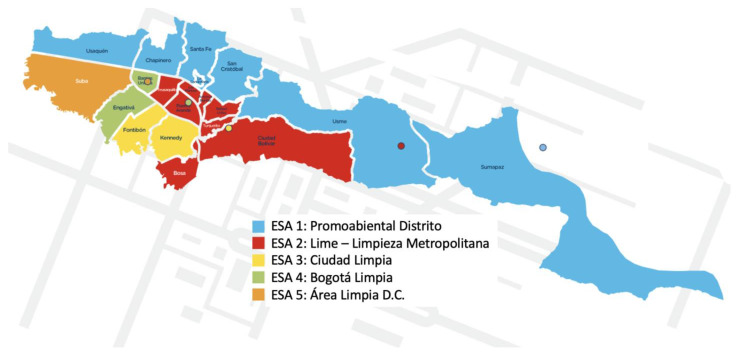
Household trash collection service providers in Bogotá [63].

**Figure 5 ijerph-20-04256-f005:**
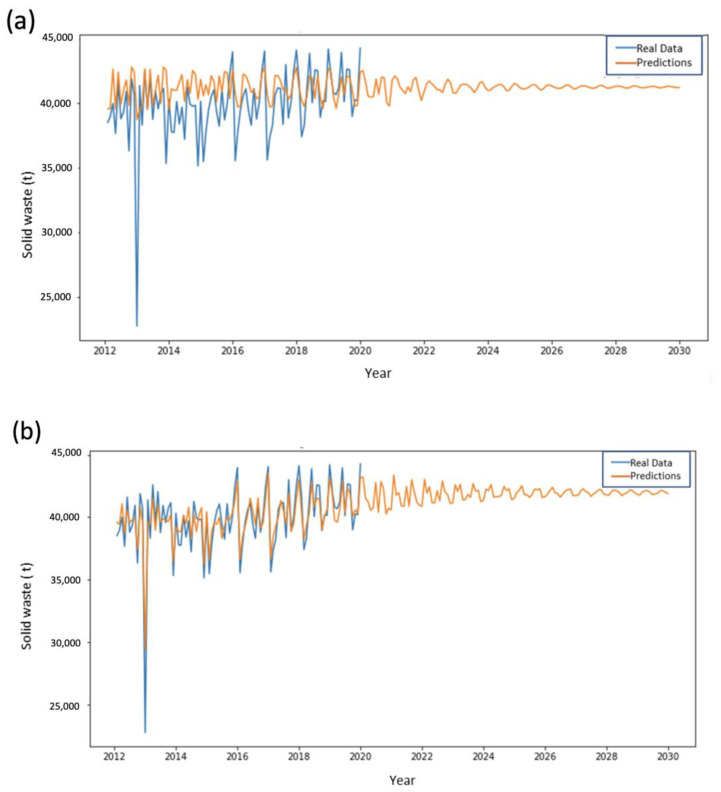
Performance comparison in the training and prediction for ESA Zone 1 using (**a**) LSTM and (**b**) SVMs.

**Figure 6 ijerph-20-04256-f006:**
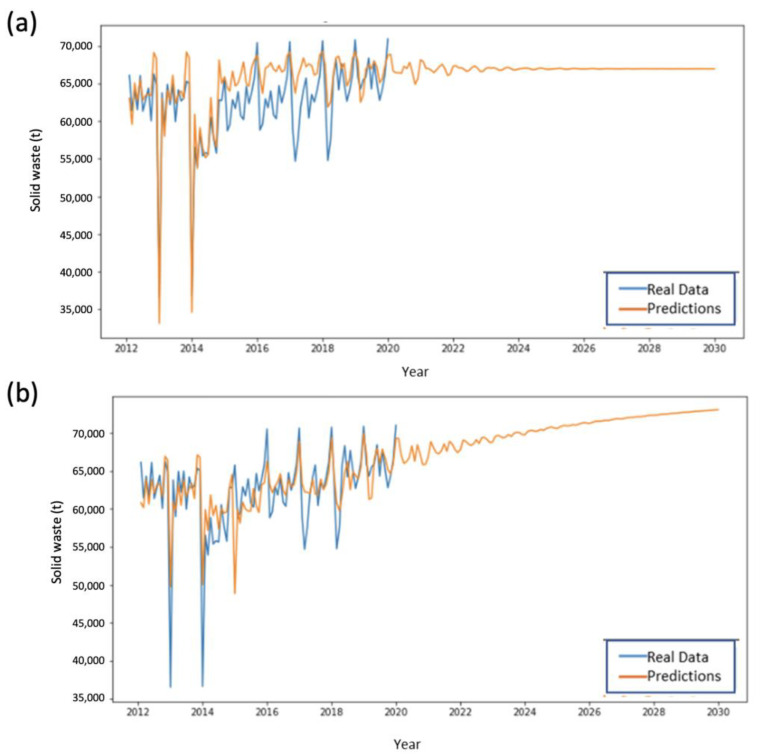
Performance comparison in the training and prediction for ESA Zone 2 using (**a**) LSTM and (**b**) SVMs.

**Figure 7 ijerph-20-04256-f007:**
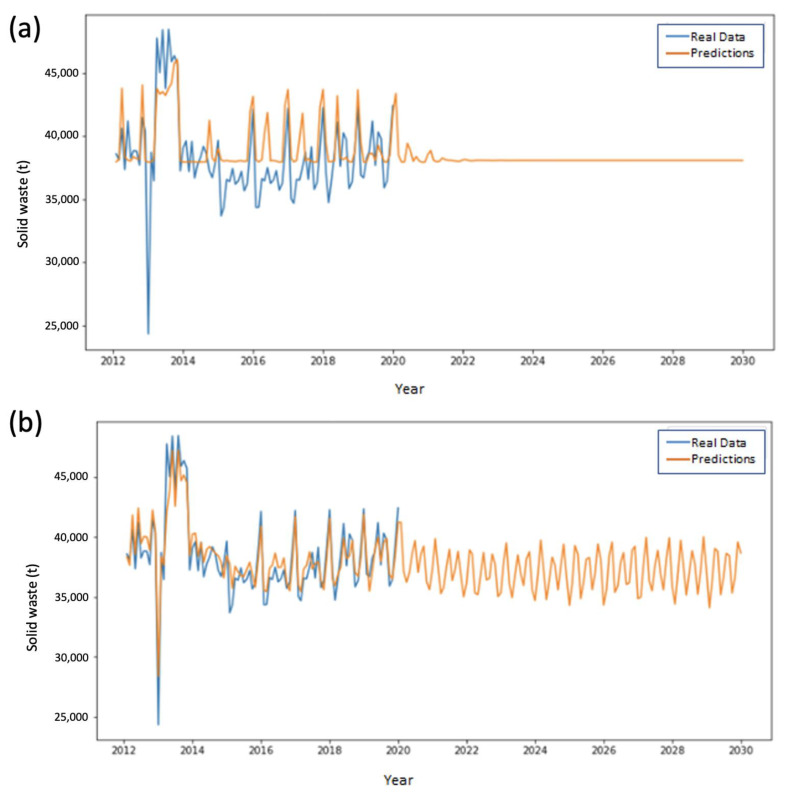
Performance comparison in the training and prediction for ESA Zone 3 using (**a**) LSTM and (**b**) SVMs.

**Figure 8 ijerph-20-04256-f008:**
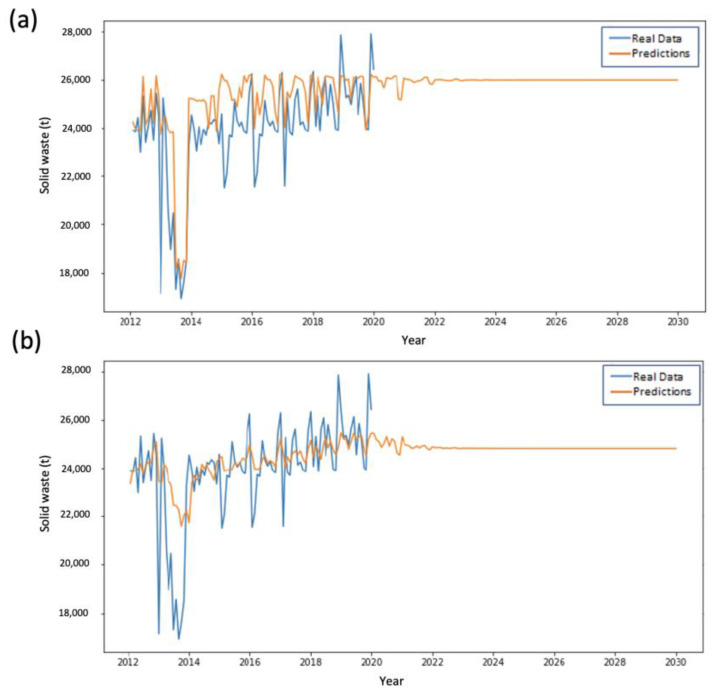
Performance comparison in the training and prediction for ESA Zone 4 using (**a**) LSTM and (**b**) SVMs.

**Figure 9 ijerph-20-04256-f009:**
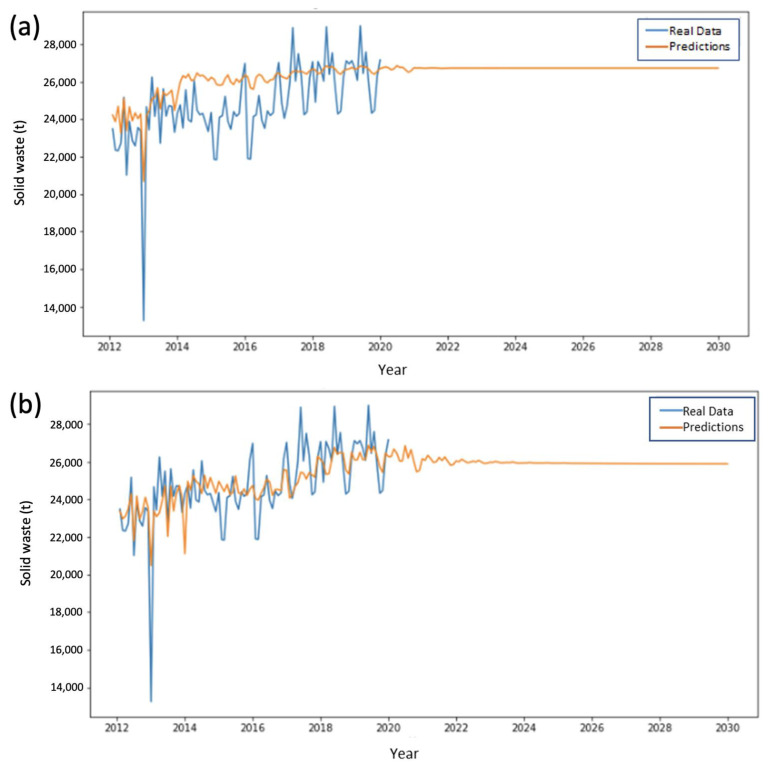
Performance comparison in the training and prediction for ESA Zone 5 using (**a**) LSTM and (**b**) SVMs.

**Figure 10 ijerph-20-04256-f010:**
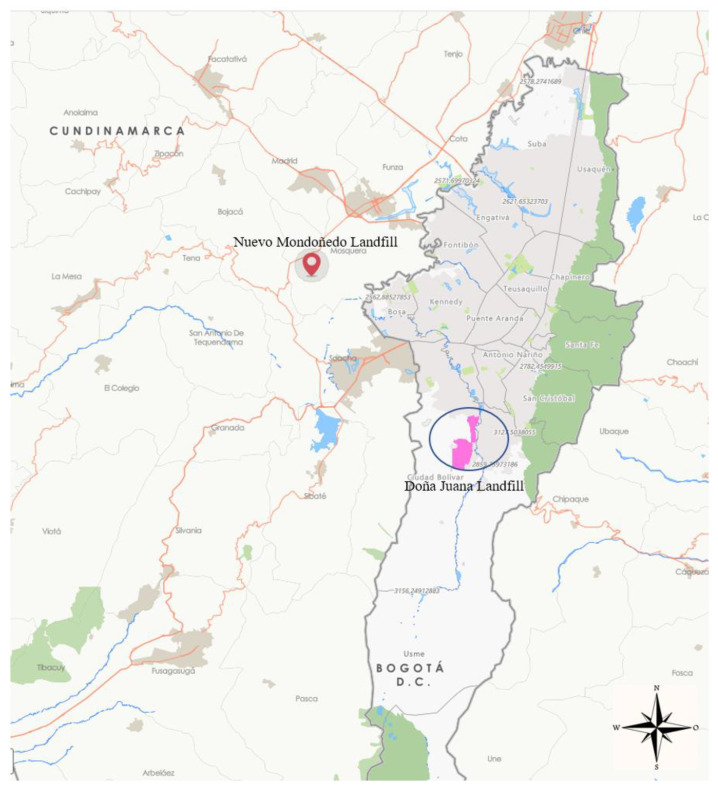
Location map of work distance 1 (Doña Juana landfill) and work distance 2 (Nuevo Mondoñedo landfill location zone) [71].

**Figure 11 ijerph-20-04256-f011:**
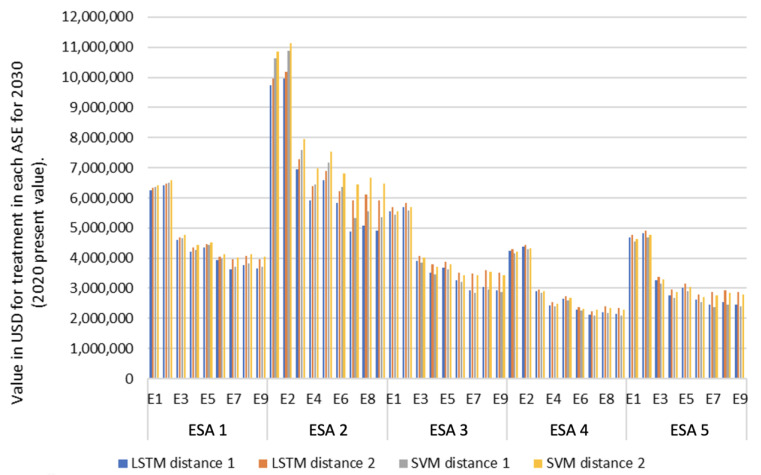
Projected costs for 2030 (monthly average) by ESA, model, scenario and distance.

**Table 1 ijerph-20-04256-t001:** Structure of the ESAs for the collection of solid waste in Bogotá [62].

ESA	Service Operator Number in Each ESA	Locality by ESA
ESA 1	PROMOAMBIENTAL	Usme, San Cristóbal, Santa Fé, La Candelaria, Chapinero, Usaquén, Sumapaz
ESA 2	LIME S.A E.S.P.	Ciudad Bolívar, Bosa, Tunjuelito, Rafael Uribe, Antonio Nariño, Puente Aranda, Teusaquillo, Los Mártires
ESA 3	CIUDAD LIMPIA	Fontibón, Kennedy
ESA 4	BOGOTÁ LIMPIA	Engativá, Barrios Unidos
ESA 5	ÁREA LIMPIA	Suba

**Table 2 ijerph-20-04256-t002:** Proposed scenarios as defined by the Inter-American Development Bank study [67].

Scenario		MSW Technology (%)
E1	Incineration	78%
	Landfilling	22%
E2	Gasification	78%
	Landfilling	22%
E3	Mechanical treatment + anaerobic digestion	68%
	Landfilling	32%
E4	Mechanical treatment+ open-air composting	52%
	Landfill	48%
E5	Mechanical treatment + closed composting	68%
	Landfilling	32%
E6	Source classification + composting	60%
	Landfilling	40%
E7	Landfilling + biogas burning	100%
E8	Landfilling + biogas energy generation	100%
E9	Landfill + biogas capture and direct sale	100%

**Table 3 ijerph-20-04256-t003:** MSE and MAE obtained by each simulation method on every ESA.

Zone	ESA 1		ESA 2		ESA 3		ESA 4		ESA 5	
Method	MSE	MAE	MSE	MAE	MSE	MAE	MSE	MAE	MSE	MAE
SVM	0.0108	0.0820	0.0419	0.1341	0.0110	0.0845	0.0946	0.1996	0.0300	0.1204
LSTM	0.0383	0.1524	0.0530	0.1800	0.0893	0.2436	0.0886	0.2204	0.0448	0.1716

**Table 4 ijerph-20-04256-t004:** Working distances considered in the modeling.

Scenario		Final Disposal Distance 1 (km)		Final Disposal Distance 2 (km)	
		ESA 4	ESA 5	ESA 4	ESA 5
E1	Incineration	27.3	36.9	27.3	36.9
	Landfilling	21.5	32.6	35.8	51.6
E2	Gasification	27.3	36.9	27.3	36.9
	Landfilling	21.5	32.6	35.8	51.6
E3	Mechanical treatment + anaerobic digestion	35.6	45.1	35.6	45.1
	Landfilling	21.5	32.6	35.8	51.6
E4	Mechanical treatment + open-air composting	35.6	45.1	35.6	45.1
	Landfill	21.5	32.6	35.8	51.6
E5	Mechanical treatment + closed composting	35.6	45.1	35.6	45.1
	Landfilling	21.5	31	35.8	51.6
E6	Source classification + composting	35.6	45.1	35.6	45.1
	Landfilling	21.5	32.6	35.8	51.6
E7	Landfilling + biogas burning	21.5	32.6	35.8	51.6
E8	Landfilling + biogas energy generation	21.5	32.6	35.8	51.6
E9	Landfill + biogas capture and direct sale	21.5	32.6	35.8	51.6

## Data Availability

The data presented in this study are available in the article and references cited.

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
