# Peer review of "Comparative Analysis of the Implementation of Support Vector Machines and Long Short-Term Memory Artificial Neural Networks in Municipal Solid Waste Management Models in Megacities"

_ijerph, 2023, doi:10.3390/ijerph20054256_

Round 1

Reviewer 1 Report

This is a very interesting article and  could have a significant contribution to its thematic area, which is related to the municipal solid waste management based on Support Vector Machines and Long Short-Term Memory ANNs. This is a well written article of good scientific quality. However, some minor modifications are needed in order to become suitable for publication.

·        Lines 64-66: ‘the most….GA” please add the associated references.

·        lines 74-75: “ANNs ….intuition” a reference is recommended for this statement

·        lines 84-85: “Furthermore..deterministic models” how do you justify this statement?

·        Lines 86-87: “ANNs are….overfitting” how do you justify this statement?

·        Introduction part: the goal of this modelling study and it’s contribution to the society must be emphasized/ clarified in the introduction part. Please add some relevant text.

·        Lines 171: figure 2: I don’t really believe that figure 2 is necessary. I recommend  to delete it.

·        Line 224: please replace the word “performed” with a more suitable one

·        Methodological model part: please add in this section some basic mathematical theory about the LSTM and SVM models you use.

·        Line 232: “3. Results and discussion”  vs line 325 “4. Discussion” you use the term discussion in both sections. Please , revise analogously.

·        Line 351: please use the abbreviation for ANNs (make sure to apply this through out the manuscript)

Author Response

Dear reviewer #1,
Thanks very much for your comments. We have carefully upgraded the
manuscript taken them all into account.
The manuscript has been reorganized and more references have been
included. While the original manuscript included 6342 words and 53 references,
this updated version includes 7936 words and 76 references.
We believe that this updated version of the manuscript fulfills all their
requirements and is now appropriate for the publication in International Journal
of Environmental Research and Public Health.
Please find below the specific actions taken to update the manuscript
accordingly to your considerations.
⁃ Lines 64-66: ‘the most….GA” please add the associated references. -
References have been added as requested
⁃ Lines 74-75: “ANNs ….intuition” a reference is recommended for this
statement
References have been added as requested
⁃ lines 84-85: “Furthermore..deterministic models” how do you justify this
statement?
A new reference [5] has been included
⁃ Lines 86-87: “ANNs are….overfitting” how do you justify this statement?
A new reference [5] has been included
⁃ Introduction part: the goal of this modelling study and it’s contribution to
the society must be emphasized/ clarified in the introduction part. Please add
some relevant text.
A new paragraph has been included inside the “Introduction” section of the
manuscript
⁃ Lines 171: figure 2: I don’t really believe that figure 2 is necessary. I
recommend  to delete it.

The figure has been deleted as requested. The content of the figure is now
explained inside the text.
⁃ Line 224: please replace the word “performed” with a more suitable one
The sentence has been rewritten as requested
⁃ Methodological model part: please add in this section some basic
mathematical theory about the LSTM and SVM models you use.
Basic theory and references have been included as requested
⁃ Line 232: “3. Results and discussion”  vs line 325 “4. Discussion” you use
the term discussion in both sections. Please , revise analogously.
This was a mistake. The section titles have been corrected.
⁃ Line 351: please use the abbreviation for ANNs (make sure to apply this
through out the manuscript)
Abbreviations have been revised through the whole manuscript

Reviewer 2 Report

The article entitled “Comparative analysis on the implementation of Support Vector Machines and Long Short-Term Memory Artificial Neural Networks in municipal solid waste management models in megacities” is well-written and, from my point of view, would be of interest for the readers of the International  Journal of Environmental Research and Public Health. In spite of this and before its publication I consider that authors should perform some changes. The changes suggested are as follows:

In the introduction authors should detail what is the purpose of the manuscript and, also, describe the structure of the article.

Line 128: please provide references for Python and C++ languages.

In the materials and methods section there is no a specific subsection about machine learning methods. From the point of view of the referee, it would be convenient to introduce in order to give to the readers some technical details and references.

Figure 3: it consists really of two figures. Please, label it as a) and b) and indicate in both of them the meaning of each curve.

The same that is said about Figure 3 can be said about Figure 4, 5, 6 and 7.

Discussion and conclusions and correct and represent properly the content of the article.

Author Response

Dear reviewer,
Thanks very much for your comments. We have carefully upgraded the
manuscript taken them all into account.
The manuscript has been reorganized and more references have been
included. While the original manuscript included 6342 words and 53 references,
this updated version includes 7936 words and 76 references.
We believe that this updated version of the manuscript fulfills all their
requirements and is now appropriate for the publication in International Journal
of Environmental Research and Public Health.
Please find below the specific actions taken to update the manuscript
accordingly to your considerations.
⁃ In the introduction authors should detail what is the purpose of the
manuscript and, also, describe the structure of the article.
The introduction section has been rewritten and the structure of the manuscript
is now included
⁃ Line 128: please provide references for Python and C++ languages.
Three new references have been included

⁃ In the materials and methods section there is no a specific subsection
about machine learning methods. From the point of view of the referee, it would
be convenient to introduce in order to give to the readers some technical details
and references.
The materials and methods sections has been expanded as requested
⁃ Figure 3: it consists really of two figures. Please, label it as a) and b) and
indicate in both of them the meaning of each curve. The same that is said about
Figure 3 can be said about Figure 4, 5, 6 and 7.
All these figures have been redone as requested
⁃ Discussion and conclusions and correct and represent properly the
content of the article.
“Discussion” and “Conclusions” sections have been revised as requested.

Round 2

Reviewer 2 Report

After the changes performed by the authors, I consider that the paper is ready for its publication. Congratulations.